# 6TiSCH on SCμM: Running a Synchronized Protocol Stack without Crystals

**DOI:** 10.3390/s20071912

**Published:** 2020-03-30

**Authors:** Tengfei Chang, Thomas Watteyne, Brad Wheeler, Filip Maksimovic, Osama Khan, Sahar Mesri, Lydia Lee, Ioana Suciu, David Burnett, Xavier Vilajosana, Kris Pister

**Affiliations:** 1EVA Team, Inria-Paris, 75012 Paris, France; 2EECS Department, UC Berkeley, Berkeley, CA 94720-1770, USA; brad.wheeler@berkeley.edu (B.W.); fil@berkeley.edu (F.M.); oukhan@berkeley.edu (O.K.); smesri@berkeley.edu (S.M.); lydia.lee@berkeley.edu (L.L.); db@berkeley.edu (D.B.); ksjp@berkeley.edu (K.P.); 3IN3, University Oberta de Catalunya, 08035 Barcelona, Spain; isuciu0@uoc.edu (I.S.); xvilajosana@uoc.edu (X.V.)

**Keywords:** crystal-free, 6TiSCH, SCμM, smart dust

## Abstract

We report the first time-synchronized protocol stack running on a crystal-free device. We use an early prototype of the Single-Chip micro Mote, SCμM, a single-chip 2 × 3 mm^2^ mote-on-a-chip, which features an ARM Cortex-M0 micro-controller and an IEEE802.15.4 radio. This prototype consists of an FPGA version of the micro-controller, connected to the SCμM chip which implements the radio front-end. We port OpenWSN, a reference implementation of a synchronized protocol stack, onto SCμM. The challenge is that SCμM has only on-chip oscillators, with no absolute time reference such as a crystal. We use two calibration steps – receiving packets via the on-chip optical receiver and RF transceiver – to initially calibrate the oscillators on SCμM so that it can send frames to an off-the-shelf IEEE802.15.4 radio. We then use a digital trimming compensation algorithm based on tick skipping to turn a 567 ppm apparent drift into a 10 ppm drift. This allows us to run a full-featured standards-compliant 6TiSCH network between one SCμM and one OpenMote. This is a step towards realizing the smart dust vision of ultra-small and cheap ubiquitous wireless devices.

## 1. Introduction

Low-power wireless networks are a key technology for applications ranging from industrial process monitoring to smart city and environmental monitoring. These networks combine time synchronization to achieve ultra-low power consumption, and channel hopping for high reliability. The resulting technology is known as Time Synchronized Channel Hopping (TSCH). TSCH is at the core of all main industrial standards, including WirelessHART [1], ISA100.11a [2] and IEEE802.15.4 [3]. 6TiSCH [4] is the latest such standardization efforts, lead by the Internet Engineering Task Force (IETF). Best-in-class commercial TSCH products today offer <50 *μ*A average current draw and over 99.999% end-to-end reliability [5].

Today, these standards can run on virtually any IEEE802.15.4-compliant chip. All of the commercial chips use stable oscillators as a time reference. A typical design consists of a printed circuit board with the main chip, and 2 crystal oscillators: a *fast* crystal (typ. 16–20 MHz) which is used to accurately select the communication frequency and clock the modulation/demodulation, and a *slow* crystal (typ. 32 kHz) used as the main timing source for the synchronized state machine of TSCH. A crystal oscillator is a small fragment of lab-grown and cut quartz, enclosed in a package, and excited by circuitry that is typically on the chip. These devices have the property of oscillating at a frequency that is precise and characterized over temperature, supply voltage, and aging. Typical drift rates, i.e., the inaccuracy of the frequency, is in the 10–30 ppm (parts-per-million) range. That is, rather than oscillating at 32768 Hz a 10 ppm crystal oscillates somewhere between 32767.672 and 32768.328 Hz. This translates to: when this crystal is used to measure a 1 s duration, it measures something between 0.999990 s and 1.000010 s, which is an acceptable error for TSCH networks.

The problem of needing a crystal is cost, space and energy. While the crystal itself might be relatively cheap (in the USD 0.50 range), using them requires one to make a printed circuit board to assemble the crystal to the chip, which consumes space and increases cost.

The promise of “crystal-free” designs is to remove the need for external crystals. Indeed, the goal of the Single-Chip micro Mote project is to remove all external components, including crystals, capacitors and other passives, and indeed ultimately even the battery and antenna, integrating everything into the wafer fabrication process. The current version of the chip still requires external power and antenna.

A related approach is to integrate the oscillating circuitry into the same package as the integrated circuit. This is what Texas Instruments has done for its recent CC2652RB: it is a System-in-Package (SiP) which combines an ARM Cortex-M4 and an IEEE802.15.4 radio on a single IC, and a separate MEMS BAW (Bulk Acoustic Wave) oscillator. The latter consists of two piezoelectric thin films, and is “*used to generate the RF carrier to eliminate the need for an external 48 MHz crystal*” [6]. This reduces design footprint and cost, and is a first step towards a crystal-free architecture, although it still consists of two technologies (CMOS and MEMS) combined into one package.

Similarly, Wiser et al. build a prototype Bluetooth Low Energy (BLE) radio which uses a thin-Film Bulk Acoustic wave Resonator (FBAR) as a replacement for a crystal oscillator [7]. To meet the ±60 ppm BLE specification on center frequency drift, they use both linear and quadratic coefficient compensation algorithms to limit the temperature effect to ±10 ppm, using an embedded temperature sensor. The remaining ±50 ppm budget is used to compensate for the effect of stress and aging on frequency error.

The single-chip design realizes the crystal-free vision entirely. The idea is to design a chip in which all oscillating circuits are inside the chip itself, and consists of different types of resonating electronic circuits (e.g. LC resonator, RC delay-based oscillator, ring oscillators). The result is a single chip which can operate without any external active components, and a key component for realizing the Smart Dust vision.

The Single Chip micro-Mote, or SCμM, is a true crystal-free chip we taped out in 2019 [8]. It is a 2×3 mm^2^ single-chip crystal-free mote-on-chip which contains an ARM Cortex-M0 micro-controller, a 2.4 GHz IEEE802.15.4 radio, and an optical receiver for optical programming. Figure 1 shows SCμM on the board we use to develop/debug it.

The Michigan Micro Mote (3M) partly realizes that vision. Known as “world’s smallest computer”, one 3M version measures only 0.04 mm^3^ and is composed of a stack of dies wire-bonded together [9]. They rely on visible light communication using an LED and a photodiode with a communication range of 15.6 cm. Another 3M mote, measuring 3×3×3 mm^3^ does include a more traditional RF transmitter [10]. Yet, because of the drift of its timing circuits, a highly-capable FPGA-based computer system is needed to receive the signals it sends, and mote-to-mote communication is not possible. A third 4×4×4 mm^3^ 3M mote [11] is capable of mote-to-mote communication, but has to rely on a crystal oscillator (assembled as one layer of the stack) for accurate timekeeping. None of these are standards-compliant, i.e., they cannot communicate with off-the-shelf radios.

The challenge with SCμM, as with any single-chip crystal-free device, is that its internal oscillators are far less accurate than crystal/MEMS-based external circuits. SCμM has a drift up to 16,000 ppm over temperature [12], three orders of magnitude higher that crystal/MEMs-based oscillators. This make it extremely difficult to *(1)* tune the frequency to communicate on, *(2)* set the correct rate to modulate/demodulate, and *(3)* keep a good sense of time to schedule communication in a TSCH network.

Why use complex networking with extremely constrained devices? TSCH does has important advantages. First, it is proven common in Industrial IoT applications, standardized in WirelessHART, ISA 100.11a and 6TiSCH, and commercialized for example in Analog Devices’ SmartMesh IP. Second, the micro-size of mote limits its energy storage capacity [13]. The synchronization based protocol, i.e., TSCH, provides ultra-low level of power consumption while mostly the other asynchronization protocol cannot satisfy. Third, SCuM was designed for TSCH, in particular its timer structure and radio interface match our OpenWSN implementation. One important point is that, once the oscillators on SCuM are calibrated for it to be able to communicate with regular motes such as the OpenMote, the same oscillators give the necessary timing to a TSCH implementation.

In [12], we showed a calibration algorithm to tune the oscillators on SCμM so it can send and receive frames to the OpenMote, a popular off-the-shelf IEEE802.15.4 mote built around the CC2538 chip [14]. In this paper, we go further and show an entire synchronized protocol stack running on SCμM. Specifically, we show that, through 3 levels of calibration and compensation, we are able to have SCμM and OpenMote drift by as little as 10 ppm, and stay synchronized with a maximum synchronization error of 300 *μ*s. Because it implements the full stack, SCμM appears as a full-featured host in an IPv6 TSCH network.

As a fact that the environment changes, such as temperature, voltage or humidity, heavily influences the RC/LC oscillator frequency error, keeping sustainable frequency error while environment changes is a big challenge. For example, the LC oscillators of SCμM drifts hundreds ppm when the temperature changes for 1 Celsius degree. The calibration and compensation techniques proposed in this article tune the oscillators to the desired frequency under constant room temperature. The goal is to express that SCμM is capable to run a full standardized protocol stack under certain circumstance.

The remainder of this paper is organized as follows. Section 2 introduces the 6TiSCH protocol stack and the OpenWSN reference implementation of that stack. Section 3 details the main features of SCμM, including its clock system. Section 4 focuses on how we calibrate the clocks to allow SCμM to communicate with OpenMote. Section 5 explains the compensation algorithm we need for porting OpenWSN onto SCμM, and presents experimental synchronization results. Finally, Section 6 summarizes this paper and discusses ongoing and future research.

## 2. 6TiSCH Stack, OpenWSN Implementation

6TiSCH [4] is a working group which is standardizing the latest protocol stack based on TSCH. It combines the industrial performance of IEEE802.15.4 TSCH, with the IETF upper stack for IoT devices. As depicted in Figure 2, this upper stack includes CoAP, UDP, RPL and 6LoWPAN.

At the core of the lower stack is IEEE802.15.4 TSCH. All nodes are synchronized to one another; time is cut into timeslots, each typically 10 ms long. All communication is orchestrated by a schedule, which indicates to each mote what to do in each slot: transmit, listen or sleep. This scheduled approach allows for ultra low-power operation, as motes only turn their radio on when they know they need to communicate with a neighbor, typically less than 1% of the time.

A pseudo-random hopping pattern is used for each transmission. The result is that, each time mote *A* sends a frame to mote *B*, it does so on a different frequency. The resulting “channel hopping” is effective at combating external interference and multi-path fading, and is also used by technologies such as Bluetooth and cellular networks.

6TiSCH builds on top of IEEE802.15.4 TSCH. Each mote in a 6TiSCH network starts with a minimal schedule [15]. The Minimal Scheduling Function [16] is used to track the amount of frames sent to a particular neighbor, and uses the 6top Protocol [17] to negotiate additional cells to that neighbor when needed. All communication is secured, and the Constrained Join Protocol [18] is used by a node to securely join a network, through mutual authentication between the network and the joining node.

OpenWSN [19] is the reference open-source implementation of 6TiSCH. It consists of two parts: the firmware running on the motes and OpenVisualizer, a Python-based application running on a PC. The firmware implements the entire 6TiSCH protocol stack; OpenVisualizer acts as the bridge between the 6TiSCH low-power wireless network and the Internet. OpenWSN has been ported to 10 hardware platforms. In this paper, we present a port of OpenWSN onto SCμM.

OpenWSN has very limited requirements for the hardware it runs on. All it needs is a single 32 kHz timer with a single compare register. As detailed in Section 3, SCμM was designed with OpenWSN is mind, and its timer structure is perfectly suited to run the OpenWSN TSCH state machine. The challenge is that SCμM has no stable time reference.

## 3. The Single Chip micro-Mote (SCμM)

SCμM is a true single-chip low-power wireless mote-on-chip which combines an ARM Cortex-M0 core, and an IEEE802.15.4 radio. It measures 2×3 mm^2^, roughly the size of a grain of rice. On top of that, it features a radio timer (RFTimer) which is designed specifically for implementing time synchronized communication protocols such as 6TiSCH (see Section 2). Loading code into the chip is done optically by using an external board which blinks an LED close to the optical receiver on SCμM  [20]. Figure 3 shows the optical bootloading process. It is single-chip by design, and replaces external clock sources (typically crystal-based) by an internal clock system described below.

SCμM operates at 1.5 V. It consumes 1.6 mW while transmitting, with an output power of −10 dBm, and 1.4 mW while receiving, with a sensitivity of −83 dBm. While the radio consumption is optimized, the same level of optimization hasn’t been implemented for the full chip, yet. We measured 150–200 *μ*A of leakage current from SRAM and analog circuits that cannot be shut off in this revision of the chip. In addition, we measure 200–250 *μ*A of current drawn by the different peripherals (including the ARM Cortex-M0 micro-controller) which use HCLK as their clock source (which cannot be turned off). This results in approximately 400 *μ*A of current.

To perform time-slotted communication with 6TiSCH, a slot with maximum length packet transmission, which takes 4.256 ms, plus the acknowledgment reception (0.8 ms) costs 4.9 *μ*C for SCμM running at 1.5 V. To receive a maximum length packet (4.256 ms), plus guard time (1 ms) and send the acknowledgment (0.8 ms), SCμM costs 4.9 *μ*C. For idle listen slot, which takes 0.4 ms, SCμM costs 0.5 *μ*C. Assuming the leakage is reduced to a reasonable level, with 1 mA current for Tx/Rx radio that turns on/off in under 100 ns, there is no doubt we can get the current below 1 *μ*A while running 6TiSCH protocol stack. We expect the next revision of the chip to implement low-power modes for the entire chip.

Figure 4 shows the clock system of SCμM. There are 4 main oscillators: three RC oscillators (64 MHz, 20 MHz, 2 MHz), one LC oscillator (2.4 GHz). A “crossbar switch” is used for routing the 4 oscillators to be used as clock sources by the rest of the chip, including the micro-controller and the RFTimer. There are 4 clocks: HCLK used as the master clock of the micro-controller, RFTimer used by the TSCH state machine, RX_CLK and TX_CLK for generating the DSSS chip rate. The crossbar switch is configured by a series of registers called the Analog Scan Chain (ASC).

Though the frequency stability of RC/LC oscillators are not comparable to the crystal oscillator, in term of combating with the influence of temperature and voltage, it is possible to calibrate the clock through software to meet the requirements.

The previous works presented in [21] shows the LC tank oscillator of SCμM drifts less than 40 ppm over 13 h in the absence of temperature changes, which meets the ±40 ppm specification of IEEE802.15.4. Over 50 °C temperature changing range, SCμM could drift over 4000 ppm of variation. By adding a feedback mechanism through the incoming packet to calibrate the frequency, the effect of the temperature variation is reduced from 150 ppm to less than 10 ppm indoor over the duration of the test, which is over 10 h, as indicated in Figure 5. Burnett et al. [22] did a thorough analysis of the stability of various clocks used in SCμM as well. For the 32 kHz RC oscillator, to keep the time offset within 1 ms, which is the minimal offset allowing two TSCH motes to communicate each other, SCμM is capable to re-synchronize every 20 s.

The RFTimer is designed specifically for TSCH. It orchestrates the transition between the different states of the TSCH state machine, as shown in Figure 6. The RFTimer comes with multiple compare registers, which allows the entire sequence of events to happen during a slot to be programmed at the beginning of the slot. The RFTimer then works hand-in-hand with the radio. For example, a frame can be loaded into the transmit buffer of the radio automatically at a specific time, without needing code to be executed on the micro-controller.

## 4. Frequency Synthesis and Clock Calibration

The goal of this paper is to show a 6TiSCH network composed of one SCμM and one OpenMote. The challenge is that SCμM does not have an accurate sense of time, and therefore derives its time reference from OpenMote. This section describes how SCμM tunes the frequency it communicates on, and how we calibrate its clocks.

We need to give SCμM a rough time reference so it can send frames that OpenMote can receive. The frequency of each of the oscillators is tunable. We designed the code running on the optical programmer board in such a way that, at the end of the bootloading process, the optical programmer repeatedly sends a sequence that causes a OPTICAL_ISR interrupt to be generated on SCμM. This interrupt fires every 100 ms for 2.5 s. While this is happening, on SCμM, all the clocks are running. By recording the counter value of each of the clocks, and knowing the interval between interrupts, SCμM calibrates each of the oscillators.

Following this coarse calibration using the optical programmer, SCμM can also calibrate against the OpenMote. We do this offline, i.e., this calibration is done once, the result of which is reused the next time SCμM is programmed. For this calibration, SCμM sends frames on channel 11 (2.405 GHz) to OpenMote. OpenMote is programmed to listen on that channel, and print over its serial port the value of its XREG_FREQEST register, which indicates the frequency offset of the incoming signal. According to that value, we manually tune the LC oscillator of SCμM, to minimize that frequency offset. The goal of this calibration is the same with what is done in [21]. The difference is that, in [21] the calibration is done on SCμM side through the intermediate frequency estimation.

This procedure is repeated for each SCμM board, as each has slightly different tuning parameters. This is illustrated in Figure 7.

While SCμM is running the 6TiSCH stack, it keeps synchronized to the OpenMote. Part of that is making sure the boundaries of its TSCH slots are aligned in time with that of OpenMote. This is done natively in the OpenWSN implementation. OpenWSN uses a 32 kHz timer with a compare value set so it fires at each slot boundary. The accuracy of the clock used by this timer influences synchronization accuracy. RFTimer runs at 500 kHz, not 32 kHz as OpenWSN assumes. This means the OpenWSN port to SCμM divides down the 500 kHz RFTimer so it appears as a 32 kHz clock source to the otherwise unmodified OpenWSN stack implementation. Since 500/32 = 15.625, the integer division applied in the port results in a rounding error. This means the slot length on SCμM is slightly different than the slot length of OpenMote. As is, this difference in slot length results in an apparent relative drift between OpenMote and SCμM. We therefore implement a digital trimming (tick skipping) compensation algorithm, detailed in Section 5.

In the implementation presented in this paper, a limitation of the FPGA/SCμM setup presented in Figure 1 is that we cannot use the 20 MHz on-chip RC oscillator (RC_20MHz in Figure 4) to source the RFTimer. Instead, we use a 20 MHz crystal oscillator of the FPGA. The results in this paper carry over to using the on-chip RC_20MHz, *except* that *(1)* the FPGA’s crystal oscillator has a much smaller drift over temperature, *(2)* the on-chip RC delay-based oscillator is expected to have higher jitter than the FPGA’s crystal oscillator. How much this impacts the overall stability of our implementation (including over temperature) when running on SCμM is left for future work.

## 5. Implementation and Experimental Results

The port of OpenWSN on SCμM has a footprint of 54 kB. This includes the full protocol stack and drivers. It takes the optical bootloader 2–3 s to load that code onto SCμM.

One of the goals of porting OpenWSN onto SCμM is to show that this platform is perfectly capable of running an off-the-shelf completely standards-based full stack. As a result, we made as little changes as possible to the OpenWSN implementation. There are, however, the following changes that we had to make.

First, the port of OpenWSN on SCμM does not come with link-layer security, nor secure joining. This is because this version of SCμM does not come with an AES-128 cipher.

Second, we were able to significantly simplify the OpenWSN TSCH state machine thanks to the RFTimer. On all other platforms OpenWSN is ported to, the time used by the TSCH state machine only has one compare register, and cannot automatically trigger radio actions. This means that, on any other platform, at the start of a transmit slot, the code schedules the timer to fire at the beginning of the TXDATAPREPARE state (see Figure 6). At that time, the micro-controller is woken up again, and loads the frame into the transmit buffer of the radio, and arms the timer again, this time to fire at the starts of the TXDATADELAY state. In contrast, on SCμM, RFTimer provides multiple timer compare registers, so the code arms the RFTimer to fire at the start of the TXDATAPREPARE, TXDATADELAY states, etc. states, all at once at the start of a slot. Moreover, using the RFCONTROLLER_REG register of SCμM  [23], the RFTimer directly triggers radio actions, significantly reducing the load on the micro-controller.

Third, we had to increase the slot length to 82 ms. This is because we are using, on this revision of SCμM, an FPGA for the digital side of the chip. Each time the FPGA configures the radio (at most twice per slot), it takes 16 ms for it to completely use the ASC. This is a limitation of the FPGA version only, which the next revision of SCμM will not have.

Because of the rounding error in clock division detailed in Section 4, the slot duration of SCμM and OpenMote are slightly different, resulting in apparent relative drift. To measure this, we program the devices to toggle a pin at the beginning of each slot. We connect both pins to a logic analyzer, and have the devices run without communicating (i.e., without resynchronizing). Figure 8 shows the evolution of the time offset between SCμM and OpenMote, over time. The drift (the slope of the line in Figure 8) is 567 ppm.

The default guard time of the OpenWSN implementation is 1 ms. This is the maximum offset between two motes, beyond which they cannot communicate. With a drift of 567 ppm, it takes less than 2 s for perfectly synchronized motes to de-synchronize beyond the guard time. Typical drift rates of crystal oscillators are in the 10–30 ppm, the apparent drift rate observed here is hence much larger, and must be compensated. To do so, we implement a digital trimming compensation algorithm. This algorithm uses “tick skipping”: it periodically adds or substracts a tick from the lengths of the slot.

As the slot communication feature of TSCH, the offset between SCμM and OpenMote is measured as the offset of their slot boundaries. Because of the frequency error, the slot length of SCμM is longer or shorter than OpenMote, which leads to the time offset between them. After a certain duration, the offset will accumulate to 1 tick (e.g. 30.5 *μ*s at 32768 Hz). Tick trimming is applied then, to extend or short the current slot length to compensate the offset.

Algorithm 1 shows how the digital trimming works in pseudo-code. $TC_{x}$ indicates the time correction in ticks at synchronizing time $T_{Sync_{x}}$. $I_{sync}$ indicates the synchronization interval in seconds. $T_{numSlots}$ indicates the synchronization interval in number of slots. Drift indicates the number of slots drifting for one tick. The digital trimming procedure is shown in the second part of the algorithm. More details that how the trimming compensation is implemented is explained in [24].
**Algorithm 1:** Digital trimming algorithm
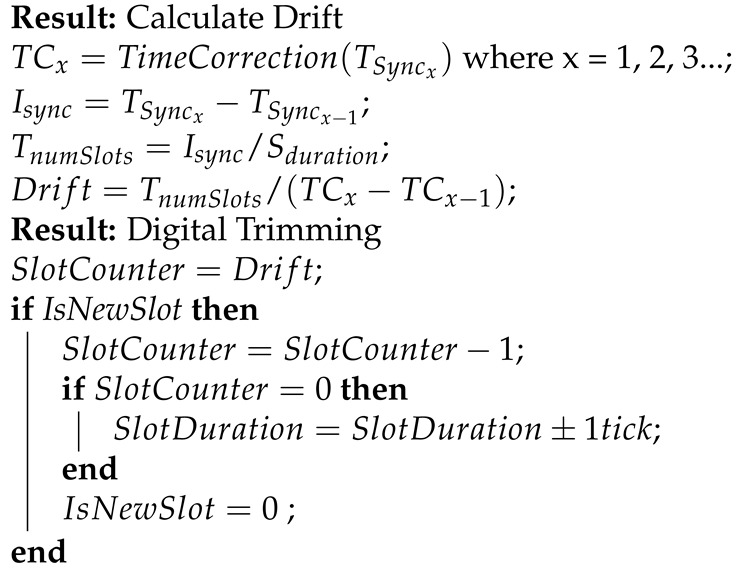


An I/O pin is toggled at the beginning of each slot on both SCμM and OpenMote side. By connecting logic analyzer to that I/O pin on OpenMote and SCμM, we calculate the time offset between SCμM and OpenMote. Figure 9 is the resulting offset when applying digital trimming. It shows two effects working together. First, the digital trimming causes the rapid saw-tooth like compensation, resulting in a much more manageable 10 ppm apparent drift. Second, SCμM regularly synchronizes to OpenMote, causing larger jumps. Overall, at constant temperature, SCμM and OpenMote remain synchronized with a synchronization error not exceeding 300 *μ*s.

This compensation allows us to build a proof-of-concept 6TiSCH network composed of one OpenMote and one SCμM. Since both implement the full protocol stack, the full functionality is available. By setting the OpenMote as dagroot, it starts to send Enhance Beacons (EB), to which SCμM synchronizes. SCμM then receives RPL DIOs packets [25], which allow SCμM to identify OpenMote as its routing parent. Using the 6TiSCH 6top protocol, SCμM reserves a transmit cell to OpenMote. We are able to issue an ICMPv6 echo request (ping) from the PC running OpenVisualizer to the IPv6 address of SCμM and see SCμM respond. To the best of our knowledge, this is the first example of a synchronized protocol stack (in this case 6TiSCH) running on a crystal-free chip.

## 6. Conclusions

This paper details the first example of a synchronized network protocol running on a crystal-free device. We use the Single Chip micro-Mote (SCμM), a state-of-the-art 2×3 mm^2^ crystal-free mote-on-a-chip, which features an ARM Cortex-M0 micro-controller and an IEEE802.15.4 radio. We use the OpenWSN protocol stack, the reference open-source implementation of 6TiSCH, a synchronized protocol stack being standardized at the IETF. The challenge is that SCμM has no stable clock source, making synchronized communication hard.

In our solution, SCμM first listens to a blinking LED to provide coarse calibration of its oscillators. Using an OpenMote, which can measure and report the frequency offset, provides a second level of more precise tuning. Finally, as SCμM and OpenMote are communicating, the OpenWSN port on SCμM uses a digital trimming compensation algorithm based on tick skipping to turn a 567 ppm apparent drift due to a rounding error into a 10 ppm apparent drift. This allows a synchronized fully functional 6TiSCH network to form between SCμM and OpenMote.

This is only a first step, with several avenues for follow-up work. First, in the platform used for this paper, the RFTimer is actually driven by the FPGA’s crystal oscillator. This, in and of itself, does not conceptually break the crystal-free nature of this work, because the frequency source used by the analog part of the chip comes from an oscillator not locked in hardware to the FPGA’s crystal oscillator. A new version of SCμM is about to be tested in which this shortcoming is lifted, and on which we can use the on-board RC oscillator to drive the RFTimer.

Second, all communication in this paper is done on a single frequency, 2.405 GHz. The 6TiSCH stack is meant for channel hopping, in which the devices hop across all 16 frequencies of the 2.4 GHz ISM band in a pseudo-random fashion. Doing so on SCμM would mean repeating the calibration on each of the frequencies, either keeping track of individual tuning parameters of each frequencies, or designing a methodology for finding one factor given that of another frequency. We have started that work, which was presented in a previously published paper [12].

Third, the network presented here is a first step only, the ultimate goal being to build a true multi-hop mesh network composed of a combination of SCμM and OpenMotes, and eventually only SCμM chips. The challenge with that is that SCμM-to-SCμM communication is significantly harder than SCμM-to-OpenMote, because of the lack of stable clock now on both devices communicating.

The promise of true crystal-free architectures is, however, enormous. They are the last stepping stone to realize the “smart dust” vision, allowing for ubiquitous, extremely small and cheap wireless devices.

## Figures and Tables

**Figure 1 sensors-20-01912-f001:**
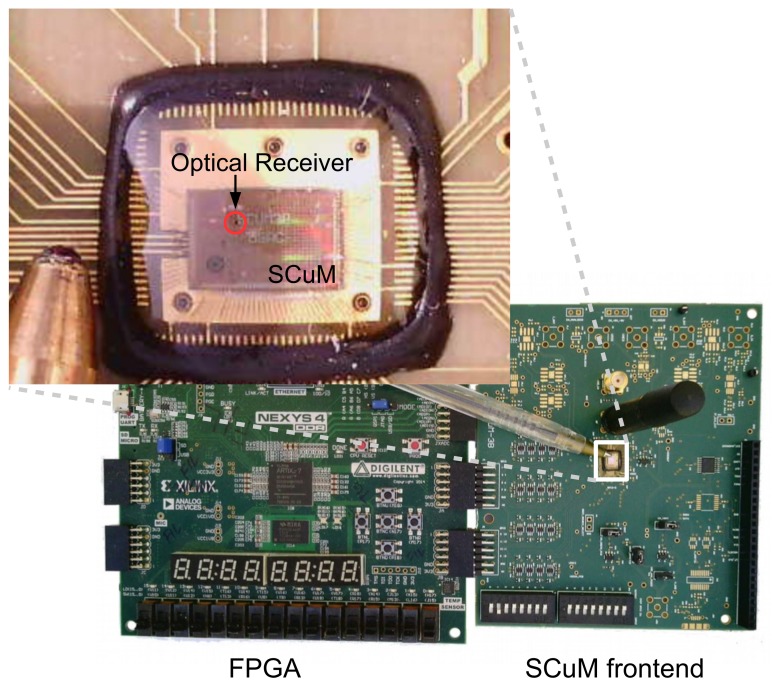
The Single Chip Micro-Mote (SCμM) is a 2×3 mm^2^ mote-on-a-chip. It features an ARM Cortex-M0 micro-controller, an IEEE802.15.4 radio, and an optical bootloader. While SCμM runs with no external components, it is shown here on its development board. In this setup, we use an FPGA board to implement the digital part (including the Cortex-M0 micro-controller), and use the analog front-end of the SCμM chip.

**Figure 2 sensors-20-01912-f002:**
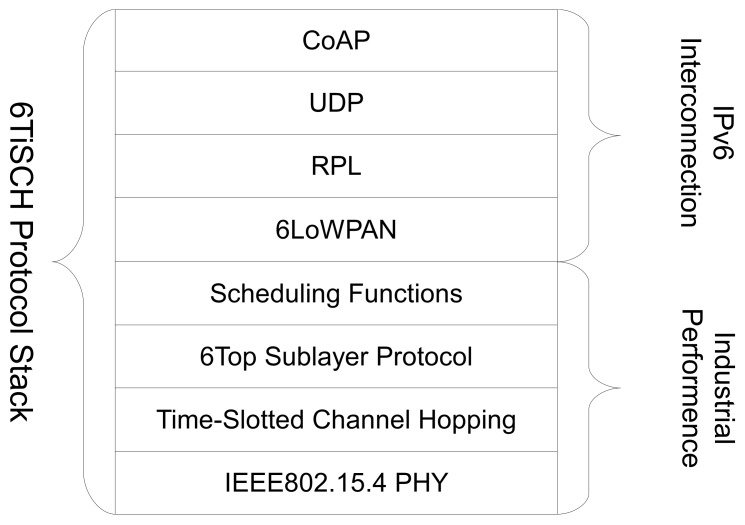
The 6TiSCH stack. The upper stack provides IPv6 connectivity. The lower stack, through IEEE802.15.4 TSCH, provides industrial-level performance.

**Figure 3 sensors-20-01912-f003:**
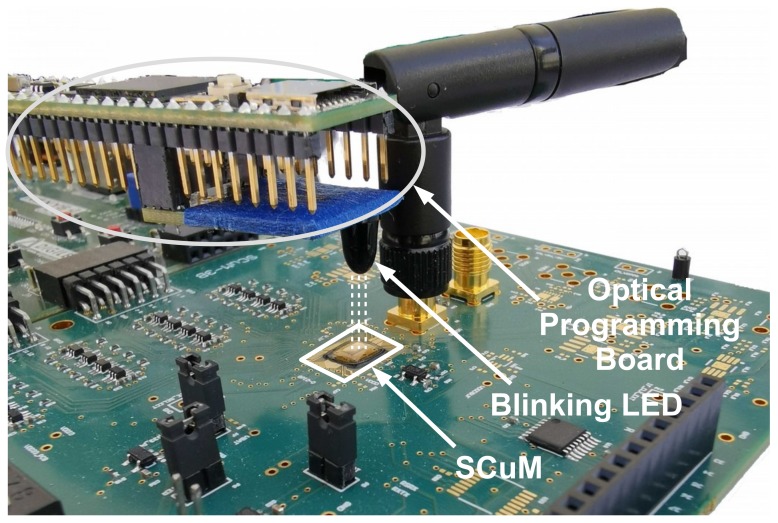
The blinking pattern of the LED on the optical programming board *(top)* is used to switch SCμM into bootloading mode and transfer the binary image to be executed onto SCμM *(bottom)* [20].

**Figure 4 sensors-20-01912-f004:**
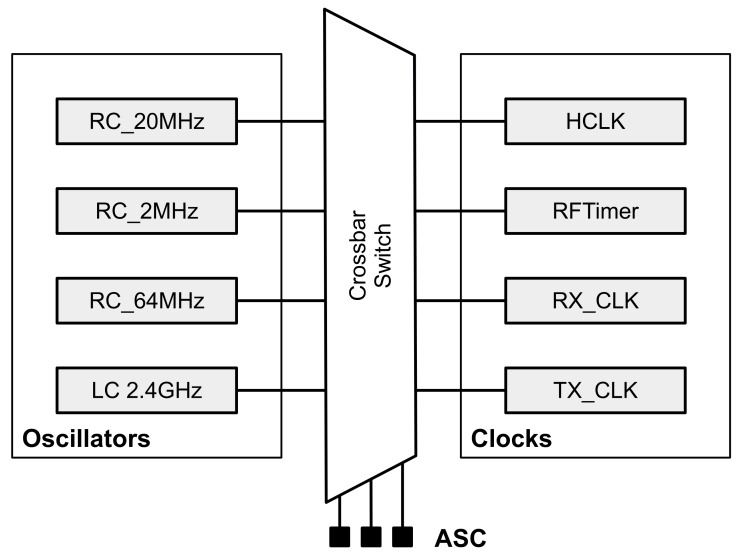
SCμM contains 4 main oscillators. A “crossbar switch” maps physical oscillators to clock sources that are used by the different peripherals in the chip. The Analog Scan Chain (ASC) is the mechanism for configuring this mapping.

**Figure 5 sensors-20-01912-f005:**
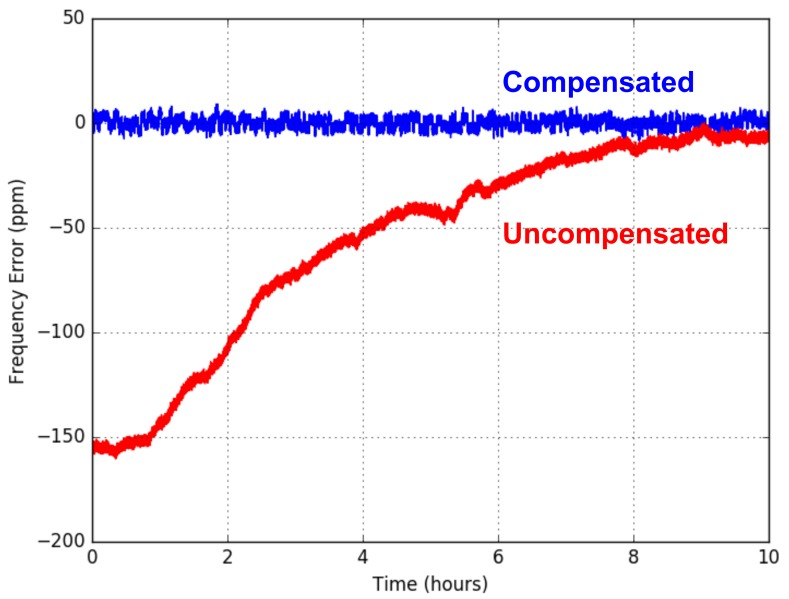
The LC oscillator frequency drift can be compensated through a feedback mechanism with incoming packets. With the feedback mechanism, the effect of the temperature variation is reduced from 150ppm (red trace) to less than 10ppm (blue trace) indoor, by testing over night [21].

**Figure 6 sensors-20-01912-f006:**
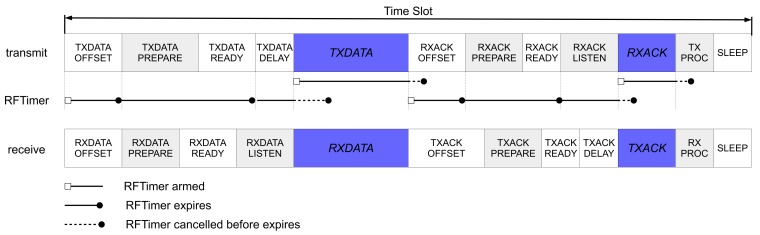
A TSCH slot is implemented as a state machine. The different states of a tramsmit and a receive slot are shown at the top. The RFTimer is used to transition from one state to the next, kicking off different actions in the radio (e.g. loading a frame in the transmit buffer) without intervention from the micro-controller.

**Figure 7 sensors-20-01912-f007:**
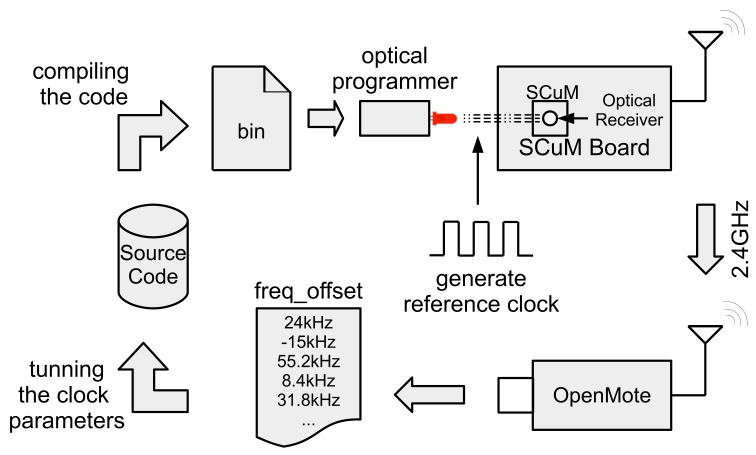
Setup used to tune the communication frequency of SCμM. SCμM transmits frames to OpenMote, which logs the frequency offset for each frame it receives. These offsets are then used to manually tune the LC oscillator of SCμM, which is used to select the transmit frequency, to minimize the mean frequency offset.

**Figure 8 sensors-20-01912-f008:**
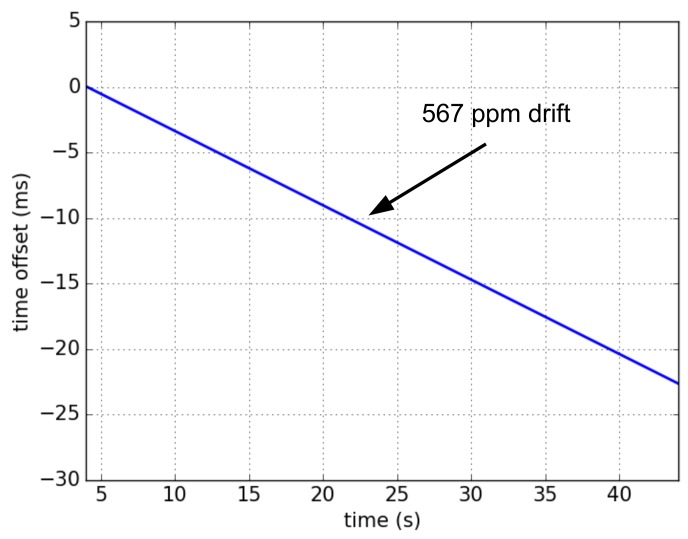
Time offset between free-running SCμM and OpenMote. No communication is taking place. SCμM emulates a 32 kHz clock source by dividing down the 500 kHz RFTimer source. The rounding error in this integer division results in an apparent drift between SCμM and OpenMote of 567 ppm.

**Figure 9 sensors-20-01912-f009:**
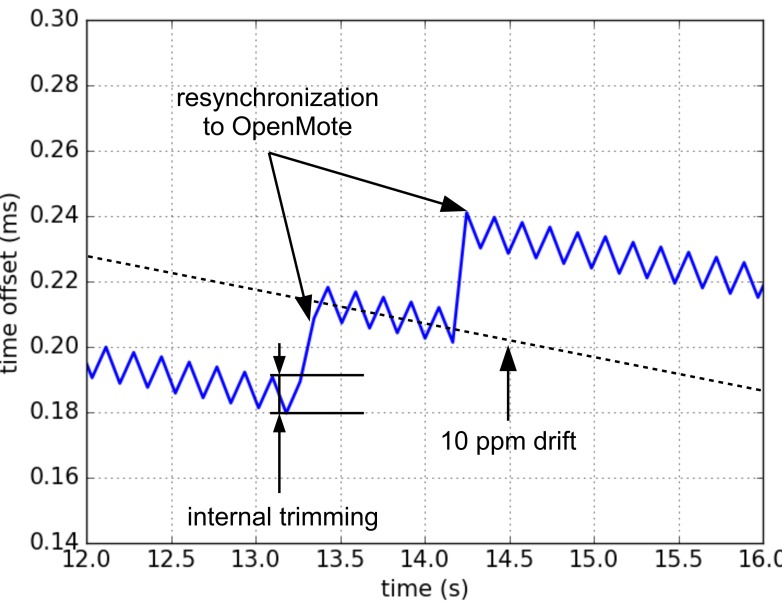
Time offset between SCμM and OpenMote when SCμM periodically re-synchronizes to OpenMote. SCμM uses a digital trimming (tick skipping) algorithm to compensate for the drift shown in Figure 8. The result is that SCμM stays synchronized to OpenMote within 300 *μ*s of synchronization error.

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
