# Peer review of "6TiSCH on SCμM: Running a Synchronized Protocol Stack without Crystals"

_sensors, 2020, doi:10.3390/s20071912_

Round 1
Reviewer 1 Report
The authors of this paper present a framework for a protocol stack, which is synchronized. The most interesting feature of the proposal is that it operates on a device without crystals. In order to do so, the authors use their own previous work of a prototype of a Single-Chip micro Mote, SCμM with an ARM Cortex-M0 microcontroller and an IEEE802.15.4 radio. Another very interesting feature of the proposal is that OpenWSN is ported for the authors' implementation. Since SCμM is only equipped with oscillators implemented on the chip, the challenge to make it work without crystals is considerable.
Two steps are carried on to calibrate the system: (1) packet reception to calibrate the SCμM oscillators, and (2) a digital trimming compensation algorithm. The first step is very interesting and it is clearly explained in the paper. The digital trimming compensation algorithm is crucial for system calibration and it is also well described. The only observation here is that for it being presented as an algorithm, it would be desirable to add at least a basic portion of pseudo-code which illustrates the algorithm to make more complete the corresponding section.
This revised version of the paper is undoubtedly better than the previous one. Just please revise the portions of the text highlighted in blue for English grammar since they need a minor spell check. Also, if possible, please include a very brief description of how the data for plots in Figures 5 and 9 can be obtained with the proposal.
Author Response
Dear Reviewer,
Thanks for the comments on the paper.
- For the digital trimming:
We have added a pseudo-code algorithm in the paper to illustrate how it works. The attachment contains the added algorithm changes.
- For the method to obtain the Fig. 5 and Fig.9 ,
Both figures are obtained utilizing the logic analyzer to connect to an I/O pin on SCuM and a reference device., i.e. OpenMote. The I/O pin is toggled at a fixed interval. In Fig.9, the interval refers to slot duration. In logic analyzer, by comparing the toggling instance of the I/O pin on SCuM and OpenMote, we can calculate the time offset between the two devices. A short description is also added in the paper before introducing Fig.9. It is available to be seen in the attachment.
Thanks again for your time to review!
Regards,
Tengfei Chang (on behalf of co-authors)

Reviewer 2 Report
- An oscillator free (without crystals) is interesting subject for battery-less sensor applications. What if the SCµM send payload data other than frequency offset, is the drift change significantly based on the size of data ? What is the relation between the size of data send each frame and drift time ?
- Is there any pseudocode/flow chart of digital trimming compensation algorithm ? In line 105 in draft article, it said will be explained in Section 5, but the detail of algorithm is not well explained.
- What if there is another new mote attached in the system, is your system perform well make the drift lower ? What is the relation if there are 1, 2 or 3 mote addition to current system to drift ?
Author Response
Dear reviewer,
Thanks for your comments to this paper.
To your question 1:
The frequency offset is reported by OpenMote, rather than SCuM. SCuM does send data packet to OpenMote. The drift of SCuM generally does not change depending on the size of data. It is mostly affected by the temperature and humidity. When sending long payload data packet, since it takes long, the final frequency offset when sending the last byte may be larger comparing to short payload packet. However, this is a common behavior of radio not specific to SCuM.
To your question 2:
Thanks for the comment! We have added a pseudo-code algorithm in the paper to illustrate how it works. The changes are shown in the attached docx file.
To your question 3:
Thanks for the interesting comment! The short answer is no. For time reference, theoretically, it should be only one. If other motes are introduced to the calibration system, they need select one among the motes as the time reference and all the devices should calibrate their own clock to that selected time reference mote. Hence, adding more mote does not reduce the drift of SCuM.
Thanks again for your time to review this paper!
Regards,
Tengfei Chang (on behalf of co-authors)

Reviewer 3 Report
The main objective of the presented work was to demonstrate an original solution of synchronized network protocol running on a crystal-free device that may a be very promising proposition in the context of wireless sensor networks, IoT and the smart dust cheap wireless applications.
The Introduction section provides sufficient description of the state of the art in the domain of wireless sensor node synchronization. Next, the single chip micro-mote solution and OpenWSN implementation were presented and method for clock calibration demonstrated, including a digital trimming compensation algorithm. At the end the experimental results were presented, showing only 567 ppm frequency drift between two motes.
The main advantage of the presented work is that it proposes a method for synchronization of a protocol stack that does not need a hardware crystal oscillator on board.
The weak point of the carried out work is that it is more focused on engineering and application side rather than on extensive research analysis supported by theoretical considerations.
The overall impression of the work is good; it does not include any significant flaws.
Below, please find some suggestions for English corrections:
Lin 52: in “algorithms to limits” use “limit”;
Line 95: replace “humanity” by “humidity”;
Line 143: in “with a output power” use “with an output power”.
Author Response
Dear Reviewer
Thanks for your comments on this paper!
We have corrected the typo errors you pointed out in a new revision. We agree that the work presented in the paper indeed focuses more on the engineering and application side. Our goal is to give a proof-of-concept in this paper to say it is possible to run a synchronized protocol without crystal. Additionally, we have added a pseudo-code algorithm in the paper to illustrate how the digital trimming works. The changes are shown in the attachment.
Thanks again for your time to review our paper!
Regards,
Tengfei Chang (on behalf of co-authors)

Round 2
Reviewer 2 Report
What is the relevance of Fig. 5 with your work ? Is your work have similar(same) result while using frequency offset feedback mechanism ? Did you do several hours running the system and measuring the frequency offset at the same time and the output is the same like Fig. 5 ?
It is better only drawing/plotting your own result unless you make comparison to another author works in single chart.
Author Response
Dear Reviewer:
Thanks for the comments!
We added that figure according to one of the other reviewers who want to know the LC OSC stability character. The figure is actually one of our previous work on SCuM.
The difference in this article is that we use a manual calibration approach to calibrate LC OSC. The figure 5 calibrates LC clock through reading the IF estimation on SCuM side during receiving. In this article, we read the frequency offset from Open Mote side by asking SCuM to transmit. This is stated in the new revision as indicated in the attachment.

This manuscript is a resubmission of an earlier submission. The following is a list of the peer review reports and author responses from that submission.
Round 1
Reviewer 1 Report
The single chip micro mote is truly miniaturized hardware chip containing oscillators and radio on an extremely small single chip and thus providing progress towards the original smart dust vision. Incorporation of IEEE 802.15.4 based radio also sets it apart from the Michigan Micro mote, which can be cited as a related work (a suggestion, given that it also incorporates optical receiver for firmware transfer).
This paper, however, is its adoption for a time synchronized protocols (particularly TSCH-based) on the crystal-free chip. Therefore, a strong justification why author choice i.e., TSCH/6TiSCH (instead of several other rather simpler wireless protocols where timing as well as frequency offset requirements are not as stringent) is made. While 6TiSCH stack is standardized by IEEE and IETF and is highlighted as one reason to do so makes some sense, but a paper can be improved if the authors spend some time talking about
Quantification of overheads associated with an additional crystal clock i.e., additional cost (sub-dollar), space, and power consumption. Why a time and frequency sensitive protocol on a crystal-free chip and not another protocol that is not as sensitive to these factors. Can other simpler protocols (such as CSMA-based) be easily implemented and smart dust vision can still be realized. Why the paper motivates multihop protocol for a hardware that is in its early days and as a first step one would likely to make a single hop communication work. The later part of the paper provides single hop experimental setup(Scum->Openmote) and no longer talks about multihop. It is therefore not clear why 6TiSCH is in the beginning. Can the scientific contribution of the paper be still complete if the focus is kept on acquiring accurate timing?
It would be good to see some evaluation (ppm drift) as environmental factors such as temperature change. Such a setup would reveal sensitivity of time synchronization to these factors.
Reviewer 2 Report
This paper describes time-synchronized protocol stack running on a crystal-free device. The approach is quite interesting although I am not sure that the cost can be really reduced.
The authors may need to address the power consumption of a crystal-free device since minimizing the power consumption is more important than minimizing the cost in many sensor applications.
Reviewer 3 Report
There are some really big red flags with the papers. On a high level, the paper does not talk about the frequency error of RC oscillators which is a function of process, voltage and temperature variation. Although the trimming with the optical receiver/programming will fix the process variation, the voltage and temperature variation hasn't been discussed in this paper which is the big challenge facing RC oscillators. Additionally, the "tick skipping" algorithm hasn't been sufficiently described. Finally, using a crystal for this operation defeats the point of this work and cannot be left for future work.